# An Internet of Things Platform Based on Microservices and Cloud Paradigms for Livestock

**DOI:** 10.3390/s21175949

**Published:** 2021-09-04

**Authors:** Jordi Mateo-Fornés, Adela Pagès-Bernaus, Lluís Miquel Plà-Aragonés, Joan Pau Castells-Gasia, Daniel Babot-Gaspa

**Affiliations:** 1Department of Computer Science and Industrial Engineering, University of Lleida, 25001 Lleida, Spain; jcg23@alumnes.udl.cat; 2Department of Business Administration, University of Lleida, 25001 Lleida, Spain; adela.pages@udl.cat; 3Department of Mathematics, University of Lleida, 25001 Lleida, Spain; lluismiquel.pla@udl.cat (L.M.P.-A.); daniel.babot@udl.cat (D.B.-G.); 4Department of Animal Science, University of Lleida, 25198 Lleida, Spain; 5AGROTECNIO-CERCA Center, 25198 Lleida, Spain

**Keywords:** sensors, Internet of Things, microservices architecture, cloud computing, precision livestock farming, smart farm, pig farming

## Abstract

With the growing adoption of the Internet of Things (IoT) technology in the agricultural sector, smart devices are becoming more prevalent. The availability of new, timely, and precise data offers a great opportunity to develop advanced analytical models. Therefore, the platform used to deliver new developments to the final user is a key enabler for adopting IoT technology. This work presents a generic design of a software platform based on the cloud and implemented using microservices to facilitate the use of predictive or prescriptive analytics under different IoT scenarios. Several technologies are combined to comply with the essential features—scalability, portability, interoperability, and usability—that the platform must consider to assist decision-making in agricultural 4.0 contexts. The platform is prepared to integrate new sensor devices, perform data operations, integrate several data sources, transfer complex statistical model developments seamlessly, and provide a user-friendly graphical interface. The proposed software architecture is implemented with open-source technologies and validated in a smart farming scenario. The growth of a batch of pigs at the fattening stage is estimated from the data provided by a level sensor installed in the silo that stores the feed from which the animals are fed. With this application, we demonstrate how farmers can monitor the weight distribution and receive alarms when high deviations happen.

## 1. Introduction

Developments in the digital era are transforming the agricultural industry by making its processes more efficient, automated, and competitive. The line that separates the virtual and the physical world is getting closer, and this approximation brings in new paradigms and challenges to become a full-fledged 4.0 industry. From the Internet of Things (IoT) perspective, the core idea is that each physical object in the real world is equipped with sensors that connect it to the virtual world. Predictive models, prescriptive models, or Artificial Intelligence (AI) algorithms can be fed with these data enriched with other context information to generate new insights to assist the decision making process. Several innovations, like sensor technology [1,2], positioning systems [3], digital image processing [4], cloud [5], and fog computing [6], among others, make this transformation possible.

Nowadays, the automatic acquisition of data, which is part of the datafication and digitization process, is undertaken in many sectors given that sensors are becoming cheaper and more energy-efficient [7]. The value of such data is directly related to the use given. The simplest use of the sensor data leads to descriptive analytics, and using the information only in this way may not exploit its full potential [8]. Therefore, given that the number of devices that can transmit data is increasing, there is a need to improve the systems for gathering, crossing, and processing the data provided by the different sources in order to support smart decisions.

To start with, several challenges arise on the data collection process: (a) raw data coming from the sensor may be inaccurate or even erroneous due to systematic and random errors; (b) some information may be partially lost due to network congestion, continuous environmental interference [9], or long distances in remote places [10]; (c) fraudulent manipulation of data by attackers affecting their veracity [11]; (d) some technical specifications from sensors manufacturers can vary the structure of their data and/or their communication interface may change [12]; or (e) huge volumes of real-time data that traditional systems cannot handle [13] may overload the systems, among others. The presence of any of these factors requires specific data operations to avoid an inadequate representation [14].

Another challenge to address is how the multiple types of sources, devices, and other information sources are connected in order to obtain a global vision. Data come from different installations, places, databases, business information systems and more, each with their own format. Computing platforms and services need to be flexible to allow diverse data exchange among internal and external tools.

With the aim to increase the value of the data collected, data scientists in the fields of Operational Research, Artificial Intelligence, or Statistics are developing advanced modeling techniques. These techniques may rely at some points in solving intensive computational processes regarding CPU (central processing unit) and memory, such as when solving optimization models or training neural networks. The management of these resources requires platforms where services can scale with resource needs.

Moreover, the human interaction with these complex techniques and smart data generated need to be presented with a user-friendly perspective, smoothing the learning curve and providing targeted information. Decision-makers need to use these tools with a natural flow to access different information levels depending on their roles and decisions.

To overcome the challenges exposed until now, in this work we propose a generic software architecture together with guidelines on how to adapt the architecture to different IoT scenarios. Therefore, the main contributions of this work is the design of a software architecture with the following properties:Support the integration of several type of sensors, in order to automatically collect real-time data and keep historical records, independently of their original format.Assist in the adoption of methodologies and mechanisms to detect and correct possible errors in order to provide the most accurate information.The capability to obtain data from other data sources such as databases or web services among others to complement the sensor data.Interoperability between internal and external systems.The provision of the needed resources to ensure the correct use of the different tools and methodologies of the digital platform.The incorporation of tools that facilitate the interaction with different complex processes and provide new insights to the users.Security mechanisms to provide the needed information and access to the different processes for the different user roles.Easy of use, implementation, management, and deployment of the architecture into different scenarios.

In order to validate this architecture design, this work presents a case study for the pig sector. A platform using this software architecture has been implemented. Data collected from feed silos equipped with a level sensor are used to predict the growth of a batch of fattening pigs. Procedures for data cleaning are integrated in order to provide data of quality to proceed with the analysis. To increase the data value, statistical models are integrated in order to estimate the feed intake and the growth of a batch of pigs. The average growth model is based on a sigmoid function from the Gompertz family which uses the accumulated feed intake. The distribution is complemented with the estimation of the variance of the weight of the animals, by using a cubic polynomial function. This information allows the farmer to infer whether the pigs are gaining weight as expected. This application also incorporates an alert system to warn the farmer if some anomaly is detected.

The rest of this paper is organized as follows. Section 2 performs a literature review of similar works. Section 3 presents the design of the cloud architecture, the technologies that have been chosen to implement it, and how it has been distributed and deployed.

Section 5 applies this cloud platform to a livestock scenario. Finally, Section 6 highlights the main conclusions of this work and future research lines.

## 2. Bibliography Review

In this section, we review the state-of-the-art of proposals found in the literature that address the design of cloud platform architectures. We limit the scope to works in the agricultural sector, which is related to our case study. We analyze if these works present IoT platforms that accomplish a set of characteristics that in our opinion are essential for IoT scenarios. These characteristics are the following:1**Decision-making assistance**: To analyze the type of tools included in each work, such as visualization proposals, the use of predictive or prescriptive analysis or the generation of alarms among others.2**Scalability**: To analyze if the platform is able to integrate new IoT devices and services to meet potentially growing demand.3**Interoperability**: To explore whether the architecture is prepared to easily communicate between different systems independently of their technology.4**Portability**: To check whether the architecture provides some mechanisms that facilitate the system to be run in different environments.

We first present a brief introduction of the works analyzed. It follows a comparison of the characteristics included.

Ferrández-Pastor et al. [15] present a platform that can acquire, process, store, and monitor data from growing cropping systems with the purpose to automate the maintenance of croplands and control the conditions that determine the proper development of a crop such as soil moisture, water pH level, or luminance. Trilles et al. [16] present the design of an architecture for IoT that manages devices, acquires data from such devices, and analyzes and generates events/alerts from these data devices. They validate the solution in vineyards to monitor environmental variables, such as the temperature, air/soil humidity, and to help the farmer predict the mildew disease.

Taneja et al. [17] present an IoT platform for animal behaviour analysis and health monitoring. Their focus is on different parameters of dairy farms to monitor the health of the cows and to detect possible anomalies.

The work presented by Cambra et al. [18] proposes an innovative IoT communication system used as a low controller irrigation system. It collects real-time data such as temperature, humidity, air, rain to monitor it, controls the actuators (components that perform mechanical actions), and detects dangers in the field. Codeluppi et al. [19] propose a low-cost, modular IoT platform to improve the management of generic farms. Authors validate the platform in a real farm to collect environmental variables related to the growth of grapes and greenhouse vegetables.

A cloud-based framework called WALLeSMART is presented in [20] which proposes an architecture to address the challenges of acquisition, processing, and visualization of a massive amount of data to assist decision-making. They focus on collecting parameters related to the dairy farms and the weather. Stevens et al. [21] present MICROCEA, an architecture developed to automate the growth of plants in urban indoor residential areas. The purpose is to monitor sensor data, such as light, temperature, and humidity, and let users program events to automate the process.

Table 1 summarizes the comparison of these works and our proposal with the main keywords of the architecture presented from a technological point of view and how they use data from sensors to assist decision-making.

All the analyzed works provide some tools to visualize data (either historical sensor data, real-time sensor data, or external services data) and perform some management tasks. Only three works present the required tools for IoT scenarios so that the data collected can provide extra information: the work in [16], where they provide models to detect diseases in the crop and generate alarms since the farmers can deal with the problem at early stages. The system developed by the authors of [18] generates alarms and create actions to deal with the adequate growth of the crops, and the solution implemented in [17] provides analysis to predict heat detection or anomalies of the cows and generate alarms to warn the farmer. Such solutions are mostly self-created platforms. However, the method in [15] uses the Ubidots IoT platform in order to provide with generic and basic visualization and analysis tools.

From a scalability point of view, all these works use the cloud computing technology in order to obtain the needed resources to integrate new services and IoT devices. However, the type of database used and how these applications are distributed must be also taken in to consideration in order for the platform to cope with the new needs of the environment. Some solutions use relational databases [18,19]. These kinds of databases are not change-tolerant as they require that the structure of the data is defined before storing it, and a small update in the schema can cause a great deal of modifications in the system that must be carefully controlled. Using NoSQL [17,20] or Time series databases [16] is a better choice for IoT scenarios due to the heterogeneous data of the sensors. For example, NoSQL databases provide flexible schemes that allow storing unstructured data without having to predefine a structure.

Another important feature is how the different applications of the architecture are distributed to ensure that it is able to support potentially growing demands. Different approaches are found: the authors of [19] propose a platform that is able to integrate new IoT devices and application modules according to the demands of farmers. The works in [15,17] take some responsibilities from the cloud such as computing and analytics in order to be performed at the edge of the network and provide better time responses. In addition, in [16,17] the authors present a microservice-based approach in order to distribute the IoT platform in a set of services that are responsible for performing specific functionalities and are independent from each other.

With regards to the interoperability, most of the studies take into consideration providing HTTP (Hyper Text Transfer Protocol) APIs (Application Programming Interfaces) to the services in order to be able to interact with the consumer applications. An HTTP API is an interface that allows the interaction of two applications by using the HTTP protocol. In addition, the most used message protocol in these works that allows a connection of the IoT devices and the cloud platform is the MQTT (Message Queueing Telemetry Transport). However, in order to integrate heterogeneous IoT devices, they might require other kinds of network protocols. In [16], the authors use RabbitMQ in order to also be able to operate with AMQP (Advanced Message Queuing Protocol) and STOMP (Streaming Text Oriented Messaging Protocol) clients.

Finally, there is also the need to provide mechanisms that facilitate the use of the architecture into different environments and scenarios. The works in [16,17,20] are the only studies that provide these facilities. In fact, they use Docker in order to package the architecture and its dependencies in order to be deployed in any environment with the advantage to select which component might need in order to adapt it to a specific IoT scenario.

After analyzing the reviewed works, we find that there is a gap in solutions that provide generic scalable, portable, and interoperable architectures that let integrate models in order to take advantage from the data collected by the sensors. It is of special importance to provide evidence in practical applications of the added value that the integration of several sources can provide in the agricultural sector, in general, and the livestock sector, in particular.

In our work, the purpose is to provide guidelines and a proof of concept to design architectures that can be reused in different IoT scenarios with the adoption of a Microservice approach, Docker, and the integration of models with a focus in Agricultural contexts.

## 3. Architecture

Basically, in an IoT scenario there are sensors that can share their data through the Internet and interoperate with cloud platforms, see Figure 1.

These sensors can give access to their data through APIs or Edge computing nodes. Usually, in most of the IoT scenarios, sensors provide APIs that allow obtaining their last reads (they are updated periodically) in semi-real-time by using the HTTP protocol. However, there are other scenarios where it is required that the response time is the lowest possible. In order to provide quicker responses, there is the need that the processing and storage capabilities of the cloud are moved near to these IoT devices, but also to use more lightweight protocols than HTTP such as MQTT for data transmission to provide real-time interaction. Edge computing nodes offer these capabilities. Nevertheless, the resources of these computing nodes are limited and in order to store historical data or perform complex tasks they must also transmit the preprocessed sensor data to the cloud. The cloud platform is the key to manage the different requirements of IoT scenarios such as huge volumes of heterogeneous data, security, interoperability, and scalability by providing the on-demand computing resources such as networks, databases, servers, storage, and others through the Internet thanks to its cloud computing infrastructure.

This generic cloud architecture (Figure 2) has been designed by taking into account the requirements presented in Section 1 but with the condition that the cloud platform must be connected with third-party APIs or Edge nodes to give response to these input data.

This cloud architecture comprises three layers (the presentation layer, the logic layer, and the data layer) and a gateway. The gateway opens the door to the cloud platform. Thus, it sits between the users and a collection of user services. The primary purpose of the presentation layer is to provide a user interface to collect the user data and display the relevant information to the user. The logic layer represents the processes to obtain the sensor data, the treatment of this heterogeneous data such as data cleaning methodologies, the generation of alarms, the execution of different algorithms to obtain valuable data for the users, and the interoperability mechanisms to interact with external systems. Finally, the data layer is in charge of storing the data that come from different sources.

These layers are implemented following a microservices architecture approach. These collections of services provide the full functionality of the cloud platform. As observed above, each service uses its technology stack and interoperates between them by using lightweight protocols and APIs. Observe that if a service needs to be updated, then only this service becomes non-operational without affecting the others. Another highlighting feature is that these decoupling limits were building the overall platform by using a unique technology. Instead, it can use the adequate technology in order to build a specific functionality.

### 3.1. Gateway

The gateway sits between the users and a collection of user services. It decouples the services and allows routing each user request to the right place, keeping track of everything, and allowing role-based access to the platform services. Finally, it adds a specific component that controls all traffic, and external interaction must pass through this component. This way reduces the complexity to add new services and components and reuse these functionalities.

The main functionality of the gateway is providing authentication and authorization mechanisms. This functionality is provided by the combination of the Shinyproxy [22] and LDAP services. Shinyproxy is an open-source software that provides the authentication and authorization mechanisms to deploy Shiny apps in a production context.

The Shinyproxy service offers a login page that sends the credentials to the LDAP directory service to check if an user exists and the roles that the user has assigned to provide the web application that it might require. This way, it provides an isolated workspace for every user session by launching a Docker container with the web application accessed by the user.

### 3.2. Presentation Layer

The presentation layer is the visible part of the cloud architecture. It represents the tool that users use to interact with the complex functionalities and models they might require. It is a user-friendly web application that can be accessed from any user device connected to the Internet. It hides the complexity of the functionalities and operations performed to obtain the desired information.

The main functionalities of this layer are gathering user input using forms and wizards, and displaying output data using dashboards and data visualization tools. This work proposes R Shiny to present the data collected by the sensors and the analysis provided by the cloud in a user-friendly manner. The data are obtained by the REST API offered by the Node js web service.

R Shiny is a package for the R language that facilitates the building of interactive web apps by providing the required components to integrate the different analysis developed in R in the web without knowing Javascript, HTML (HyperText Markup Language), and CSS (Cascading Style Sheets). It also provides the mechanisms to interoperate with external systems such as APIs to obtain and transmit the required information. Therefore, it permits agile deployment and maintenance for technical and non-technical users.

### 3.3. Logic Layer

This layer aims to process the data from the presentation layer, the sensors, and external services through defined business operations and query the data layer. It acts as the bridge that allows communication between the presentation layer, the IoT devices, the external services, and the data layer. APIs allow interoperating between the different components of the architecture independently of the technology stack that has been used. It also lets the integration of new components to provide more data, such as more sensors, external services, and others. Most of the time, these APIs use the REST (Representational State Transfer) principles to operate respecting the HTTP protocol and transfer the required resources by using data formats such as JSON (JavaScript Object Notation) and XML (Extensible Markup Language). Finally, different scripts have been developed to perform data collection and data analysis to extract the value of this data and generate alarms. Thus, the main functionalities of this layer are as follows:**Data Collection**: Collect data periodically and store them into the MongoDB database.**Data Cleaning**: Detect and correct data errors, outliers, or erroneous input data.**Data processing and Analysis**: Execute predictive and prescriptive models. Transforming raw data into valuable data.**Alert Notification**: Inform the users via email about possible problems.

Most of these actions and functions must be executed periodically, such as data collection, data cleaning, and data processing. Thus, CRON jobs are implemented to automate these executions periodically and make the platform act autonomous. The technologies that have been applied to implement these functionalities are:Node js [23]. Node js is an application runtime environment that allows writing JavaScript for web applications. It comes with many adequate libraries for back-end development, such as file system management, HTTP streams, or database management. It has been chosen to develop the web service and implement an alert notification system since it facilitates dealing with multiple client requests and provides mechanisms that help scale the applications. The authors propose using Node js to implement the APIs because they have previous know-how and successful projects implemented on it. However, it is possible to choose similar technologies such as Falcon, Asp.Net, or Spring to develop these APIs. The web service is in charge of retrieving the data from the database and receiving the client’s data to process the data and store it. These data are transmitted and received using the JSON format. The communication between the client and the web service is made via the REST API with the HTTP protocol that defines the different CRUD (Create, Read, Update, Delete) operations that different controllers define. These controllers can operate with the NoSQL database by using a library called mongoose that permits mapping the models defined by Schemas into collections of the database. The alert notification system uses the nodemailer library to send emails to the user when the system detects an alert.Python. In order to perform the data collection process, Python has been chosen. It provides the mechanisms to perform HTTP calls, MQTT protocol, and query NoSQL databases. A Python script has been implemented to collect the sensor data provided by an External API and uses a package called pymongo to store the data into the NoSQL database. In order to integrate new sensors, it would be necessary to develop the corresponding script using the protocol that would be required for that sensor. Note that Python is one of the most powerful scripting languages, but other languages can be used to implement these services. The authors suggest using Python because it has a large community and several data mining, automation, and big data platforms rely on it. Moreover, it is versatile, flexible, easy to manage and maintain, and it has a vast range of libraries available.R. In order to perform the data cleaning and analysis, R scripts have been developed. These scripts interact with the Web service to store the cleaned data and store the outputs of the analysis methods. It is also possible to implement these services with other programming technologies. Nevertheless, the authors suggest using R to simplify the integration with the presentation layer and to allow operations research experts to develop models and analytics, as R is a language broadly used in this research field.

The methodology presented relies on sensors with data transmission capabilities such as HTTP or MQTT. Therefore, the only requirement to add a new sensor to the platform is defining the rules to transmit the data using the APIs. Nevertheless, this is not a limitation, and if the sensors are old-fashioned and do not support these protocols or they do not have direct access to the net, a simple solution is to add a file submission microservice to the platform. Then, the manager of the sensor can manually submit the readings periodically to the platform.

### 3.4. Data Layer

The data layer is in charge of storing data from the sensors, user profiles, user roles, and data coming from external services, among others. This layer needs to be write-optimized to handle all the data arriving from the devices. Another important feature is that data are alive and can change in the future, so the database technology must permit the addition of new data types or change the current structure. Therefore, NoSQL is most proper for this purpose than relational databases.

MongoDB [24] is an open-source NoSQL database oriented to documents and is write-optimized. It allows storing unstructured data as a document in a representation called BSON (Binary JSON) in a collection of documents. It provides a dynamic schema that lets us build flexible models by updating the schema without affecting the other documents. These features allow adapting quickly to changes that can be vital when the system starts to store huge volumes of data. This database is also designed to support horizontal scalability achieved by adding new machines and is managed by the cloud computing technology and provides high-performance to perform simple operations. Finally, it also saves costs and complexity in comparison to relational databases. These features make this database ideal for IoT scenarios.

On the other hand, OpenLDAP [25] is an open-source implementation of LDAP that allows access to directory services. These directories are specialized databases optimized for reading, browsing, searching, and simple update operations. These features make these databases ideal for providing centralized user administration to store sensitive information and other users’ account details. Furthermore, to make more accessible the management of this LDAP directory, an administration user interface is provided. LAM provides a user-friendly interface to manage an LDAP directory without the need for the administrator to manage LDAP entries.

## 4. Deployment

This section explains how the software architecture has been deployed by leveraging the advantages of Virtual Machines and container virtualization using Docker.

Virtual Machines are digital representations of physical computers with their Operative System and assigned resources (Memory, CPU, storage, and networks) created from a software component called a hypervisor. This virtualization allows the cloud infrastructure to be split up by independent Virtual Machines that can act as different servers by sharing the resources managed by the hypervisor. Moreover, Docker can be installed in these Virtual Machines [26]. Docker is an open platform that allows splitting up the applications from the infrastructure into Docker containers and dealing with process isolation. Docker containers allow separating service and dependencies from the underlying operating system and other containers to avoid dependency conflicts. All these containerized applications share a unique operating system, so it permits saving resources and it is also faster to be started and stopped than Virtual Machines that need each one an Operative System.

The cloud platform chosen to develop this work is OpenNebula. This is an enterprise-ready platform that helps build an Elastic Private Cloud. It avoids risks and vendor lock-in by choosing a powerful but easy-to-use, open-source solution. OpenNebula is based on virtual machines but also allows containerized applications from Docker to be run.

OpenNebula offers an open and transparent means to build private clouds. The Stormy server [27] is a legacy private cloud platform supported by public funding, developed and maintained by the authors’ research group aimed at assisting researchers and deployed with OpenNebula and KVM. There are several manuscripts, such as that in [28] or in [29], that use the Stormy service as a support platform to achieve their goals. Nevertheless, this architecture is not limited to this cloud provider as it has been given the needed mechanisms to be reused in different environments such as Amazon Web Services (AWS) or Google Cloud.

Concretely, we have instantiated different virtual machines using a Centos7 OS image to host the different microservices. Each virtual machine is protected with a firewall. All the microservices are deployed and managed using container-based virtualization (Docker). This containerization allows this architecture to be scaled and reused so it can replace the services used in this architecture for the specific services and its suitable technologies to implement the specific IoT scenario and also the environment where these services are deployed but also duplicate the services that require more availability. Therefore, it facilitates that this cloud architecture can be reused in any environment and is prepared to take advantage of autoscaling policies provided by Kubernetes or other cloud orchestrators.

Figure 3 depicts a general-purpose deployment into a cloud provider (OpenNebula, Amazon Web Services or Google Cloud). First, it is configured the virtual resources into the cloud provider (CPU, Memory and Networks). Then, the operating system (Centos7 in our deployment) is installed to each virtual machine deployed together with all the required tools (Docker and Git). Next, an orchestrator is installed and configured. Our deployment uses docker-compose, but other alternatives that can be used are Kubernetes, docker-swarm, among others. In parallel, the development environment can be configured in a local machine to implement and test the services and the applications. Then, to integrate the development and production environments, GitHub is used (push and pull actions). Finally, the *docker build* commands are employed to create the images, and to run and execute all the services the *docker-compose up* commands are used.

## 5. Case Study

The rate of adoption of sensor technologies varies among sectors, and it is related to the trade-off between the costs and potential benefits. In particular, the agricultural sector uses different sensors to measure and track the evolution of several environmental characteristics such as soil humidity, air temperature or CO_2_ concentrations, among others. This type of information is relevant to guide the decisions on the best actuation at each moment. The data analysis that stems from this information usually falls in the descriptive area. With the IoT platform presented, we aim to facilitate the development, usage and adoption of applications involving advanced modeling techniques for the agricultural sector. In this way, new developments from the Operations Research or the Artificial Intelligence communities such as prescriptive optimization models [30], predictive statistical models [31] or classification models [32], among others may be easily transferred to the final users.

The current case study applies the described cloud architecture to build a smart application that estimates the growth of pigs based on the amount of feed discharged from the silo, where a sensor is installed. Pork is one of the most consumed meats in the world together with poultry [33]. It is estimated that between 60% and 70% of the total production cost per kilogram of pork corresponds to feed consumption [34], being the information related to the feed a key characteristic to monitor and control. This application has been developed within a demonstration project with the participation of three farmers and a total of eight silos monitored and sixteen batches analyzed (composed of 500–900 animals per batch). The capacity of the silos monitored ranged from 11,000 kg to 16,000 kg.

The interest in estimating (and better understanding) the growth of a batch of pigs is one of the key breakthroughs that the pig sector is working on. Some approaches go in the line of precision feeding where individual data on feed intake and weight is monitored each time an animal visits the feeding station [35]. While this system offers accurate and comprehensive information, it is still an expensive option. At the other end, traditional pig management software offers estimates on the expected growth based on estimates from previous batches. An estimation of the feed consumed is built from these parameters and then compared with the loads performed during the growth. However, the interest of monitoring the feed dynamics is increasing and other solutions are being implemented. The application we have developed leverages the information provided by a low-cost sensor installed in a silo to infer the feed consumption and from this information compute the expected growth.

The primary use of a sensor installed in a silo is to provide information on the amount of feed available to guide the farmer on the replenishment order. However, this information can provide extra value if it is combined with other information. For example, feed manufacturers may better plan their operations if they know the inventory levels of their clients [36]. From a farmer’s perspective, the enrichment of statistical growth models with sensor data that estimate the feed intake may provide valuable estimates of the expected growth evolution of a batch of pigs. Monitoring the animals’ growth is relevant to support the optimal moment for delivering the pigs to the abattoir and to detect abnormal deviations between the theoretical expected growth curve and the estimated growth based on sensor data. Such deviations may reveal the occurrence of some (sanitary, feed quality, etc.) problem if the estimated curve is slower or reflect the result of a good practice if the growth is faster.

In the following sections, the application that estimates the weight evolution of the pigs in a fattening farm based on the estimate of the feed consumption is presented. We first explain the process to integrate several sources of data with the predictive models, the characteristics of the farm, the sensor technology used, the growth models, and the user-friendly interface that allows to easily obtain the information. The resulting application can be accessed from https://gcd007.udl.cat/login (version 31 August 2021) and the functionalities can be explored with a demonstration test set for a user (username: demo) with password (demo@alba25.udl.cat). Note that some functionalities are disabled for this demonstration user.

### 5.1. Process Description

In a three-site system, pig production is divided in three stages: (a) maternity, where the piglets are born and stay there for 4–5 weeks; (b) rearing, where young pigs are grown up to the age of 8–10 weeks; and (c) fattening, where the pigs are grown for 17–20 weeks until they reach a target marketing weight and then are sent to the abattoir. In this application, our target is to estimate the growth at the fattening stage. Figure 4 shows the main steps of the data flow process, from the data acquisition to the final growth estimate. The data sources are the sensor measures and some technical parameters of the batch of pigs which are combined to provide an estimate of the average feed consumption per animal. The literature provides functions that relate the average consumption of pigs with the expected growth. The expected average weight is finally modeled as a weight distribution, so the farmer has a week by week estimation of the number of animals that are expected to be in each weight range.

Due to sanitary reasons, the animals at the fattening stage are grown in batches where no new animals can enter the farm before the last animal from the previous batch has left and the farm is sanitized. However, the entrance and removal of animals can happen on different days. The application shows information for a particular batch, which is defined with the earliest entrance of animals. For each entrance, the date, number of piglets, and the average weight needs to be specified by the farmer in an input form. This is the basic information the farmer obtains for each truck of piglets received.

At the entrance, piglets may not have a homogeneous weight nor age. Furthermore, the growth rate of each animal will be different and therefore some animals will reach the target weight earlier than others. When a sufficient number of animals (which is related to the truck capacity) reach the marketing weight, those animals are sent to the abattoir. It is customary that the marketing window comprises four or five weeks. When a group of animals is sent to the abattoir, the farmer is expected to inform of the number of animals removed and the weight reported by the abattoir. Having the information on the number of animals present in the farm is very relevant so the application can compute the average individual feed intake more accurately.

#### 5.1.1. Feed Consumption Estimation

The driver information for estimating the amount (in kg) of feed consumption of the batch is provided by the sensor. By monitoring the silo level, we can infer the amount of feed that the batch is consuming. We assume that the amount of feed discharged from the silo is consumed by the animals disregarding any lost it may happen due to rejection or waste. There exists several solutions in the market for remotely monitoring a silo contents. A direct approach to know the silo’s content weight is to install load cells in the silo support structure [37]. This is the most precise approach but for already installed silos, it demands a costly installation. An indirect approach is to use level or surface sensors, which are based on radar, laser or other type of guided wave technology which provides information on the volume of the silo that is occupied. Therefore, the weight of the feed in a silo needs to be approximated by combining the measure of the sensor with the feed density. These type of sensors are usually placed at the top of the silo and require minimal installation. They are usually powered by a solar cell which removes the need to be connected to a power grid.

In this work, the sensor used is a wireless one-point level sensor developed by Monitoring Control Systems [38] based on laser technology. In order to compute the level, it takes into account the height of the silo and it uses a time-of-flight computation to a specific point, usually the center of the silo. Then, it measures the time that the laser beam (Class II laser, <1 mW, 635 nm) uses to go to this specific point and to come back (see Figure 5). This type of sensor is appropriate for small and medium bins with up to a range of 40 m, such as those used to store feed for animals; otherwise, it may suffer from lack of accuracy. The reading is done periodically; in this work, an automated CRON job is executed every two hours. This reading is sent through a radio network of wide coverage and low electricity consumption (Sigfox network [39]) to the company platform called Digitplan. We access this information through a REST API in order obtain the silo data via HTTP protocol.

The information provided by the sensor is therefore a measure of volume which needs to be transformed to weight (see Figure 6). For doing so, we need to know the density of the feed which is different for each truckload that refills the silo. This characteristic demands to control for the points in which the reading increases due to the feed refilling operation. To reduce inaccuracies, it is expected that the farmer introduces the real weight of the added quantity. The system assumes a default density, and by comparing the estimation of the amount introduced according to the default value and the real value provided, a more precise estimation of the density can be done. The weight of the contents of the silo is estimated by multiplying the occupied volume of the silo times the density of the feed.

To approximate the amount of feed supplied to the batch would suffice to subtract two consecutive readings expressed as weight. However, the process of discharging the feed from the silo leaves the surface uneven making the readings fluctuate among consecutive periods (instead of being monotonically decreasing within any two refilling operations). Due to this reason, data cleaning is done by means of a centered moving average. Punctual missing data are also interpolated using this procedure. Figure 7 illustrates some of the data cleaning procedures that the application automatically performs. The dots show the days and the amount in which the silo was refilled. By default, the application assumes a feed density, but the density is re-estimated with the information about the real weight of the refilling load. Then, the series of the weight with the refilling-based density is computed and is cleaned in order to remove sensor noise or missing values.

There exists other sensors on the market that could be used instead of the sensor presented [40,41]. Each sensor can use different technologies (radar, 3D level sensor, etc.) to provide information on the filled volume and each manufacturer provide access to the collected data in a specific way. The platform allows to include different scripts to import data such that the different sensors can easily be included. Other solutions directly weight the feed unloaded [42] which eliminates the need to perform the transformation from volume to weight, and only the data cleaning step should be performed.

Once the point reading is cleaned, the accumulated weekly consumption is estimated (see Figure 8). This estimated consumption based on the sensor readings is compared with a theoretical accumulated feed consumption which is computed according to technical parameters that reflects usual patterns of the farm and the breed of animals that hosts. These parameters are provided by the user and are the average daily gain (ADG), the expected feed conversion rate (FCR), and the age (in weeks). Among the alternatives to estimate the expected accumulated feed intake (AFI) for a batch of *N* animals at week *t* [43], the exponential function is used:(1)$$AFI_{t}=N\frac{A}{1+\exp-(b\cdot age_{t})}$$
where *A* is the expected amount of feed taken at adult age and *b* is a fitted constant.

#### 5.1.2. Growth Estimation

Several empirical models exist to estimate the growth curve for the average population by estimating the average weight along the time. However, the average weight of a group of pigs is comprised of individuals with differing performance levels and individual growth curves. Empirically is observed that as live weight increases, weight variation between the pigs also increases. Therefore, from a farmer’s point of view, the distribution of the weight conveys better information. In this work, a growth curve from the Gompertz family [44] has been used to estimate the average weight (AW) of the population at a particular week *t*:(2)$$AW_{t}=A\exp^{-\exp(b-k\cdot AFI_{t})}$$
where $AW_{t}$ is the average body weight (kg) at week *t*, *A* is the asymptotic adult body weight (kg), *b* is an empirically fitted parameter that makes the starting point flexible, *k* is an empirically fitted parameter related to the rate of growth, and $AFI_{t}$ is the average accumulated feed intake (kg) up to week *t*. Given that a batch may be composed by subgroups of animals, one for each entry, Equation (Equation 2) is applied to each subgroup. Figure 9 shows the expected weight for each entry of the batch at a specific point in time, computed with the average feed intake according to the theoretical function and also with the estimated data from the sensor. This information has commercial significance, as deviations between those curves alert the farmer to review the animals in order to find the cause of the deviance. However, before diving into the causes, data accuracy has to be verified. When a significant deviation between the theoretical and sensor-based estimations is detected, an alert message is sent to the farmer to verify that the data corresponding to the weight of the silo’s refilling information have been provided.

The distribution of the weight at a particular point in time is represented by a Normal distribution with the mean according to Equation (Equation 2) and variance computed from a cubic polynomial function. To provide a better picture of the weight of the animals in the batch, all the subgroups distributions are merged to compose the final weight distribution. Figure 10 shows the weight distribution at a particular week with the expected number of animals within a 5 kg range. This information is especially relevant to plan the delivery of animals to the abattoir based on a target weight.

#### 5.1.3. Data Input Process

The reading and data processing of the sensor data are automated within the application. Other information related to the batch is required to be provided by the farmer. In its current state of development, the user must perform a manual entry. This is general information to set up a batch, inventory information about the number of animals present in the farm either entrances or departures, and the information on the refilling truck loads. The reader is invited to explore this information with the demo user provided in the application.

The application also includes a descriptive (and simplified) view to monitor the available feed in the silo. It shows an already processed range of historical data corrected by the density estimates and it is compared with the cleaned series.

### 5.2. Deployment

Figure 11 shows the deployment of the architecture explained in Section 4 applied to the case study. This proof of concept is deployed using two virtual machines (VM1 and VM2) under OpenNebula cloud provider. VM1 contains the gateway and is the open-door to the internet. VM2 is only accessible from VM1. As we commented, we use Centos7 as operating system in both virtual machines. This figure also highlights all the services that are started using Docker compose orchestrator (green and yellow boxes). The yellow boxes represent the services related to the data layer, and the green ones represent the APIs, scripts, and other services discussed. Note that the data collection service is gathering data from the sensor every to 2 h and stores it into the database using HTTP communication protocol. Next, the data cleaning service processes incoming data and checks for errors and non-consistent data. In parallel, the alarm service checks information from the database to inform the user, if needed. Finally, the RShinny Application uses farm service (API) and the growth model to interact using HTTP and HTTPS communication protocols to display information to the final user (client). This communication is controlled (authenticated and authorized) through the gateway.

## 6. Conclusions and Future Work

The paper concludes by arguing the current need for designing architectures that support evolving and emerging sensors and devices to facilitate digital transformation in agricultural contexts. The findings of this study can be understood as general guidelines of features to be taken into account when designing Information Technology solutions to assist agriculture. These guidelines are more consistent with current research showing that most features and requirements are standard in other industrial sectors.

Nevertheless, in this study, we shed light on the challenges and issues of applying these methodologies and Fog, Edge, and Cloud paradigms to agricultural scenarios.

One of these requirements is to deal with the heterogeneous data of sensors and external services to connect the physical and virtual worlds. Broadly translated, the state-of-the-art findings indicate that this requirement can be solved by using Fog and Edge paradigms or developing scripts in the cloud side (as we propose) that can be worth operating with these devices by using the corresponding protocol such as HTTP and MQTT and storing the data into NoSQL databases such as MongoDB.

Furthermore, another important characteristic that must provide the cloud platform is scalability. We have shown how a modular design using a microservice approach helps in adjusting the computational resources on demand. Therefore, we show how container technologies such as Docker can integrate more sensors, analysis methods, and services, and provide horizontal scaling.

Together, the present study confirms that Docker technology also facilitates the portability and interoperability of the platform to other IoT scenarios and Cloud providers by only installing this technology into the cloud. Moreover, there is also the need to provide security mechanisms such as roles and authentication to provide access to the needed tools and prevent undesired access to the platform.

The case study shows a practical implementation of the ideas presented in this work where all the architecture and the application is developed with open-source initiatives. The application shows a way to enhance the value of the data gathered by a sensor level in a silo from where a batch of pigs are fattened. While special care has been taken in the design of a general and flexible architecture so that it can receive data from other sensor manufacturers, much attention has also been paid to the integration of advanced modeling tools, such as the growth model. Note that the code developed at the modeling stage with the R statistical software was easily transferred to the platform development team, allowing a fluid flow of work. Although the application is focused to the livestock sector, the conceptual architecture presented is general and applicable to other agro-industrial processes. We leave as a future work, performing a proof of concept of the methodologies in other settings.

Finally, regarding future work, we are interested in integrating new kinds of sensors to analyze critical tasks for the case study that can be assigned to Edge Nodes and provide new analysis to provide valuable output. Functionality that would be interesting to integrate would be registering IoT devices to the platform by using, for example, QR (Quick Response) codes and the configuration/management of these devices to control proprietary devices. From a user perspective, having a push notification complement would enhance the reception of alarms. This and other characteristics may be included in future developments. Another exciting characteristic for IoT scenarios is product traceability that can be implemented by leveraging Blockchain technologies. Furthermore, to improve the analysis performance, a processing distributed framework such as Apache Spark could be integrated to provide support for R and Python programming to provide complex analysis with huge amounts of data.

## Figures and Tables

**Figure 1 sensors-21-05949-f001:**
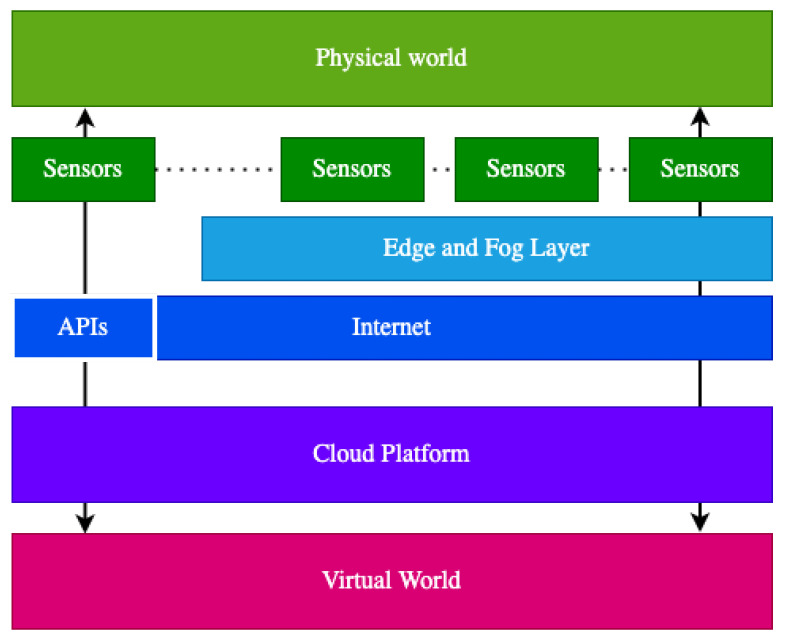
General Internet of Things (IoT) scenario.

**Figure 2 sensors-21-05949-f002:**
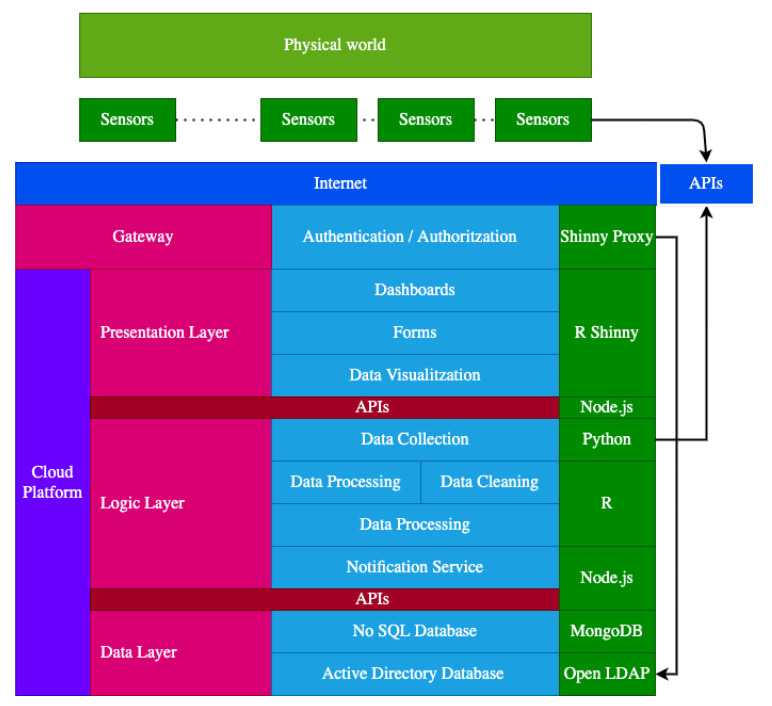
Cloud architecture design.

**Figure 3 sensors-21-05949-f003:**
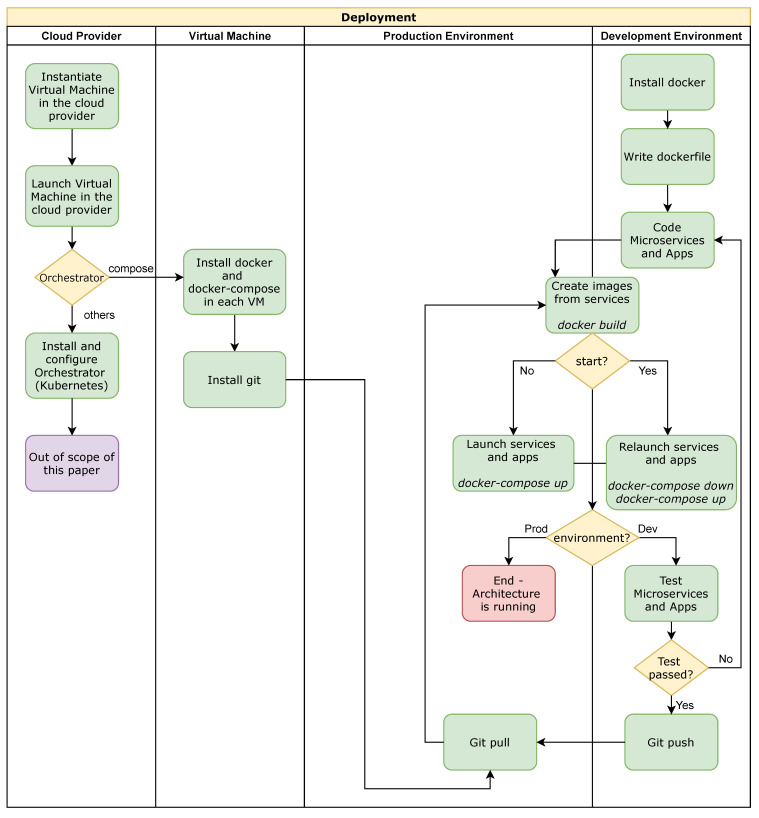
Block diagram of the deployment of the architecture.

**Figure 4 sensors-21-05949-f004:**
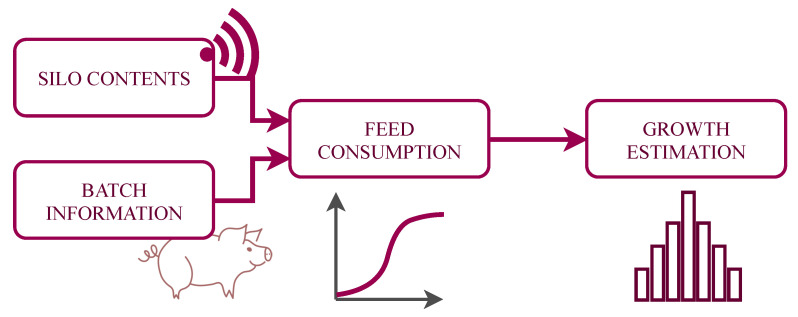
General data flow process.

**Figure 5 sensors-21-05949-f005:**
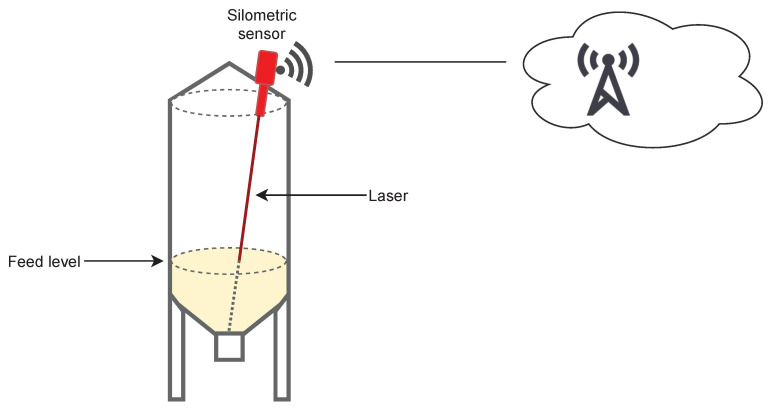
Diagram of the spatial distribution of the sensor in the silo. Technical details can be found on the manufacturer web page [38].

**Figure 6 sensors-21-05949-f006:**
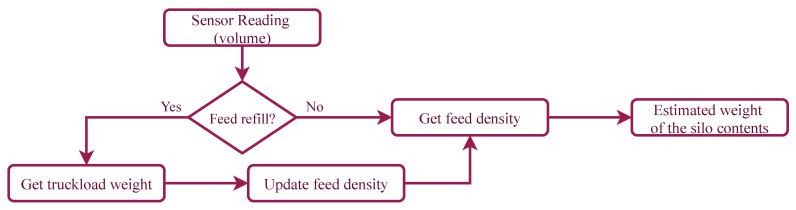
Estimation of the weight of the feed from the readings of the sensor.

**Figure 7 sensors-21-05949-f007:**
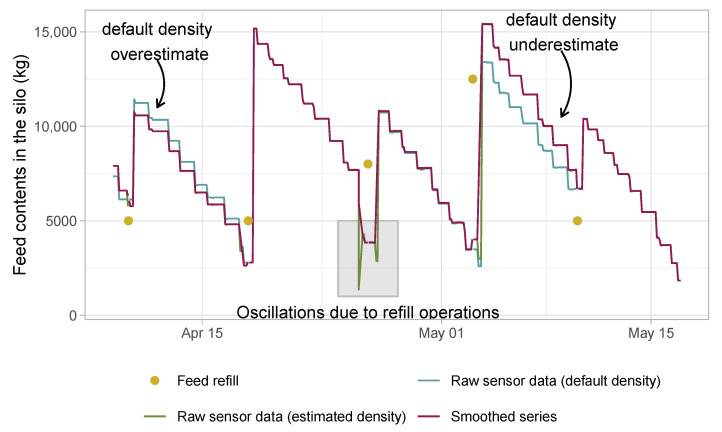
Time series of the silo contents estimated from the sensor readings.

**Figure 8 sensors-21-05949-f008:**
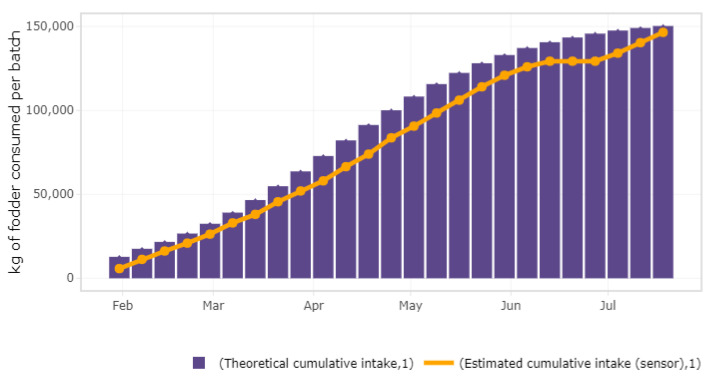
Cumulative feed intake of the animals in the batch.

**Figure 9 sensors-21-05949-f009:**
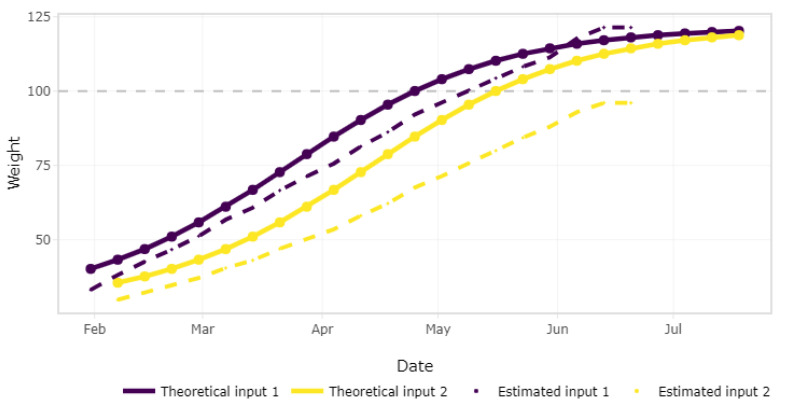
Animal average weight.

**Figure 10 sensors-21-05949-f010:**
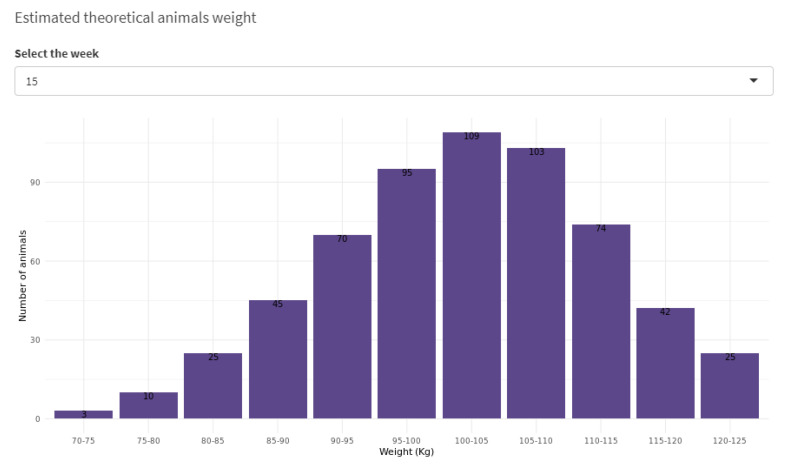
Animal weight distribution in a specific week.

**Figure 11 sensors-21-05949-f011:**
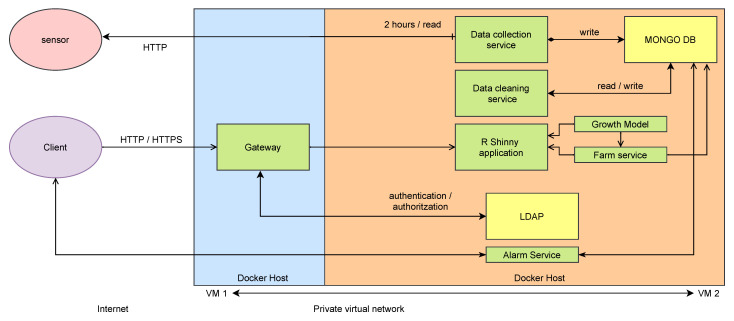
General scheme of the deployment of the case study.

**Table 1 sensors-21-05949-t001:** Comparison between related studies.

Proposal	Platform keywords	Decision Making	Scalability	Portability	Interoperability	Sector
[15]	Cloud Computing Edge Computing	Descriptive Analytics	✓	✓	-	Agriculture
[16]	Cloud Computing Docker Microservices Time Series DB	Descriptive Analytic Predictive Analytic	✓	✓	✓	Agriculture
[17]	Cloud Computing Docker Fog Computing Microservices No SQL databases	Descriptive Analytic Predictive Analytic	✓	✓	✓	Livestock
[18]	Cloud Computing Relational Databases	Descriptive Analytic Predictive Analytic Prescriptive Analytic	-	-	-	Agriculture
[19]	Cloud Computing Relational Databases	Descriptive Analytics	-	-	-	Agriculture
[20]	Cloud Computing Docker No SQL and relational databases	Descriptive Analytics	✓	-	✓	Livestock
[21]	Cloud Computing No SQL databases	Descriptive Analysis	✓	-	-	Agriculture
Our proposal	Cloud Computing Docker Microservices NoSQL databases	Descriptive Analytic Predictive Analytic	✓	✓	✓	Livestock

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
