# Peer review of "An Internet of Things Platform Based on Microservices and Cloud Paradigms for Livestock"

_sensors, 2021, doi:10.3390/s21175949_

Round 1

Reviewer 1 Report

The paper is well written, provides an excellent introduction and related works. The proposed architecture is updated, easy to implement in the real world and well described. The presented technologies deal with a good number of languages, databases and applications. In my opinion is a paper ready to be published, without any changes.

Author Response

Thank you very much for your positive evaluation of our work.

Reviewer 2 Report

The article is interesting and fits into the directions of development of IoT. The concept presented has great potential and deserves attention. Unfortunately, this is just a concept. The authors could provide a class of devices that would allow the solution to be run. They too could explain how to adapt existing devices, e.g. I have an old and efficient silo with analog sensors - how to add such a silo to the system?

The literature review identified similar solutions. There are no references to commercial solutions. It is also worth getting acquainted with the so-called gray literature, e.g. online forums and blogs. The IoT community is very active (Arduino, ESP8266/32, Raspberry).

The work devoted a great deal of space to discussing the software solutions used. The information presented can be easily found on the Internet. The justification for choosing such solutions has been omitted. The authors write, for example, "No SQL databases are an excellent choice for storing heterogeneous data such", but why do you think it is excellent for this purpose? To say so, you should compare different solutions and choose excellent one. You can realize your system in a .NET environment. .NET native should be more efficient than Python. MS SQL Server can be hosted on docker or Azure (which is even more scalable). The problem of different data structures can be solved by using a data warehouse and appropriate transformations. The authors should justify the choice of such technologies and indicate the advantages of this choice.

Minor error - row 164: advanteg.

Author Response

Please find our answers in the attached document.

Reviewer 3 Report

Here are some comments resulting from the review of the submitted manuscript:

-> The information placed in the abstract does not adequately describe the work developed. It is not very clear if a new communication platform based on IOT and cloud is proposed or if a new methodology for construction of this type of technological systems is presented. Additionally, the summary must make clear the type of innovation that the research offers so that the reader can make comparisons with other existing investigations. Also, it is important in the abstract to describe in general terms, the results obtained, i.e., if the implemented platform presented good results making it clear that it is a novel result for the area.

-> In line 52, the authors do not specify the type of architecture proposed. Is it a communication architecture? Software architecture? Hardware architecture? Hardware and software architecture for communication?

-> Lines 71 and 72 do not specify the types of statistical models used, this information is important for readers.

->In Section 2 (Bibliography review) the characterization and comparative analysis of the research works carried out by the authors is robust.

-> In line 169, the authors affirm that their proposal is innovative, when compared with the others in terms of the use or integration of models. This fact is a very weak justification since many researchs and commercial applications use advanced and intelligent mathematical models , as well as models that can evolve as the inputs and parameters are updated. In this sense, it is important that the authors analyze their work and correctly describe the degree of innovation that their proposal can bring for the specific application.

-> In Section 3, the  exposed in detail the concepts of a robust and flexible general architecture to integrate different types of sensors. In theory this architecture should work for different applications related to livestock processes and, personally I think it can be used for different agroindustrial processes.

-> Line 297 "Deployment", this subsection explains the methodology used for the implementation of the architecture, the numbering (3.5) creates confusion with the sequence of the explanation of the architecture components. Once the information in this section is important, it should be placed in a separate section.  Additionally, the information described in this section is of great importance for the application of the architecture. Therefore, for better understanding, the explanations should be illustrated by means of block diagrams or execution diagrams.

-> In line 337 the authors make the following statement: "With the IoT platform presented, we aim to facilitate the development, usage and adoption of applications involving advanced modeling techniques for the Agriculture sector." Could the authors specify what types of advanced models and what avanced techniques are applied or used?

-> In Section 4,  there is a lack of relevant information. It is important to remember that this manuscript is being submitted to a scientific journal Sensors. Therefore, it is important to make a detailed description of the technical characteristics of the sensors, the place where they are placed, their operation, the conditioning of signals used, ranges of maximum and minimum values, communication protocols used, etc.. From the point of view of technical formalism and instrumentation, a diagram (hardware architecture) of the spatial distribution of the instrumentation, devices and systems used, as well as the type of communication used (industrial networks) must be placed.

-> The authors do not provide a detailed explanation of the experiment carried out (case study, Section 4), i.e., size of the silo, location of the sensor in the silo, maximum and minimum storage volume of the silo, among other information. This information is relevant to determine the degree of detail in which the proposed architecture was tested.

-> In lines 423 and 424, it must be specified if the parameters ADG, FCR and age are placed as system inputs? Will the system calculate it? Or are they simply used as comparison parameters?

-> In Eq. 1, place within parentheses the calculated argument in the exponential function (exp (---)). Also, in the same equation there is a confusion between Age and age in the denominator of the equation.

-> The resolution of the images in Figures 6 and 7 should be improved. Additionally, the axle units in Figure 7 must be placed.

-> In line 455 the authors make the following statement: "The reading and data processing of the sensor data is automated within the application". This statement is not supported by the information placed in this manuscript, since technical details are not included, as well as the hardware architecture used in the work is not explained in depth.

-> The following conclusion is made by the authors (lines 490, 491 y 492): "While special care has been put in designing a general procedure to receive the data from any sensor manufacturer, great attention has been put into the integration of advanced modeling tools, such as the growth model. " The authors did not carry out tests with other models of sensors from other manufacturers, this kind of conclusion cannot be affirmed from the results obtained.

Author Response

(The authors gave the same response as above.)

Reviewer 4 Report

Dear authors, kindly check specific comments on manuscript structure of manuscript throughout and suggested to modified changes for the improvement of manuscript.

  • Line 1: Author should recheck the abstract section and add the statement for finding/results of the research work.
  • Line 1-10: Write the specific objective of your study in the abstract.
  • Line 1: author suggested structuring the manuscript as per the MDPI research article format.
  • Line 28: Check the statement or revised it?
  • Line330: how many farms were chosen for the case study?
  • Line345: also mention which other sensors you are using in this study.
  • Line 374: Write the details process for new animals added to the farm for feeding and how it can be integrated into the existing system.
  • Line 384: What is the role of IoT in Feed consumption estimation?
  • Line395: Write the full form of the MC System.
  • Line 364: author suggested adding some future work for developing a mobile app.

Author Response

(The authors gave the same response as above.)

Round 2

Reviewer 2 Report

Thank you for your response. The most important remarks have been corrected. 

Author Response

Thank you very much for your comments and quick review.

Reviewer 3 Report

I agree with the alterations, corrections and comments made by the authors during the review process. There are some small errors that were detected in this new version of the manuscript:

-> Line 375,  the reference to Figure, i.e.  "depicts a general-purpose ....", this reference should appear before the Figure appears in the manuscript.

-> Lines 357-366, to avoid the use of the first person (we/ I) in the manuscript text.

Author Response

Thank you very much for your comments. We have updated the manuscript properly correcting the issues you have detected. You can find the modifications in blue colour (idem to last time).